# Multidimensional responses of grassland stability to eutrophication

Qingqing Chen [1,2], Shaopeng Wang [1] ✉, Elizabeth T. Borer [3], Jonathan D. Bakker [4], Eric W. Seabloom [3], W. Stanley Harpole[2,5,6], Nico Eisenhauer [2,7], Ylva Lekberg [8], Yvonne M. Buckley [9], Jane A. Catford [10], Christiane Roscher [2,5], Ian Donohue[9], Sally A. Power [11], Pedro Daleo[12], Anne Ebeling[13], Johannes M. H. Knops[14], Jason P. Martina[15], Anu Eskelinen [2,5,16], John W. Morgan[17], Anita C. Risch [18], Maria C. Caldeira [19], Miguel N. Bugalho [20], Risto Virtanen [16], Isabel C. Barrio [21], Yujie Niu [22], Anke Jentsch [22], Carly J. Stevens [23], Daniel S. Gruner [24], Andrew S. MacDougall[25], Juan Alberti [12] & Yann Hautier [26]

Eutrophication usually impacts grassland biodiversity, community composition, and biomass production, but its impact on the stability of these community aspects is unclear. One challenge is that stability has many facets that can be tightly correlated (low dimensionality) or highly disparate (high dimensionality). Using standardized experiments in 55 grassland sites from a globally distributed experiment (NutNet), we quantify the effects of nutrient addition on five facets of stability (temporal invariability, resistance during dry and wet growing seasons, recovery after dry and wet growing seasons), measured on three community aspects (aboveground biomass, community composition, and species richness). Nutrient addition reduces the temporal invariability and resistance of species richness and community composition during dry and wet growing seasons, but does not affect those of biomass. Different stability measures are largely uncorrelated under both ambient and eutrophic conditions, indicating consistently high dimensionality. Harnessing the dimensionality of ecological stability provides insights for predicting grassland responses to global environmental change.

In 2020, the Convention on Biological Diversity reported that only 8% of the world's nations met the target of limiting excess nutrients to a level that is not detrimental to ecosystem functioning[1]. This failure means that eutrophication, which disrupts biodiversity, functionality of many ecosystems, including grasslands, and nature's contributions to humanity[2,3], could threaten our long-term survival and prosperity. Meanwhile, climate extremes (e.g., droughts and floods) are increasing in both intensity and frequency, which can also have severe negative impacts on our society and ecosystems[2,3]. Whether, and how, effects of eutrophication propagate to affect ecosystem stability in the context of increasing climatic variability remain elusive.

In ecological studies, stability is a multifaceted concept that characterizes the ability of an ecosystem to minimize fluctuations in its properties against perturbations and variations in environmental conditions[4]. Traditionally, ecosystem stability has been assessed through temporal invariability, which involves calculating the mean of an ecosystem property divided by its standard deviation[4–6]. As a result, temporal invariability has been commonly referred to as temporal stability. In this paper, as we consider multiple facets of stability (including invariability) to quantify ecosystem responses through time, we use invariability to avoid confusion. Ecologists have increasingly recognized the importance of measuring stability using

resistance during, and recovery from, climate extremes such as droughts and floods[7] (Fig. 1). However, most studies have focused on individual stability facets, particularly temporal invariability and/or resistance to drought[8,9]. A few studies (mostly single-site experiments) exploring multiple facets of stability have showed that different stability facets are often uncorrelated, and these correlations or lack thereof may differ under global change factors including climate extremes, eutrophication, consumer removal, light addition, and heatwaves[8,10–16]. Strongly correlated stability facets essentially represent a single dimension, which means that understanding the mechanisms behind one stability facet can provide crucial information for understanding and predicting other facets of stability. For example, if two stability facets are positively correlated, manipulating ecological factors that boost one stability facet should lead to an increase in the other. If two stability facets are negatively correlated, improving one facet of stability will likely come at the expense of the other, making it difficult to optimize both stability facets simultaneously. When different facets of stability are uncorrelated, they effectively represent a high dimensionality and the mechanisms that govern each stability facet are likely to be different[8].

Stability can be measured for multiple community aspects such as aboveground biomass, community composition, and species richness (Fig. 1) and the stability of different community aspects may correlate with one another[10,17,18]. While most studies have focused on the stability of aboveground biomass, the stability of other community aspects such as community composition and species richness may be essential

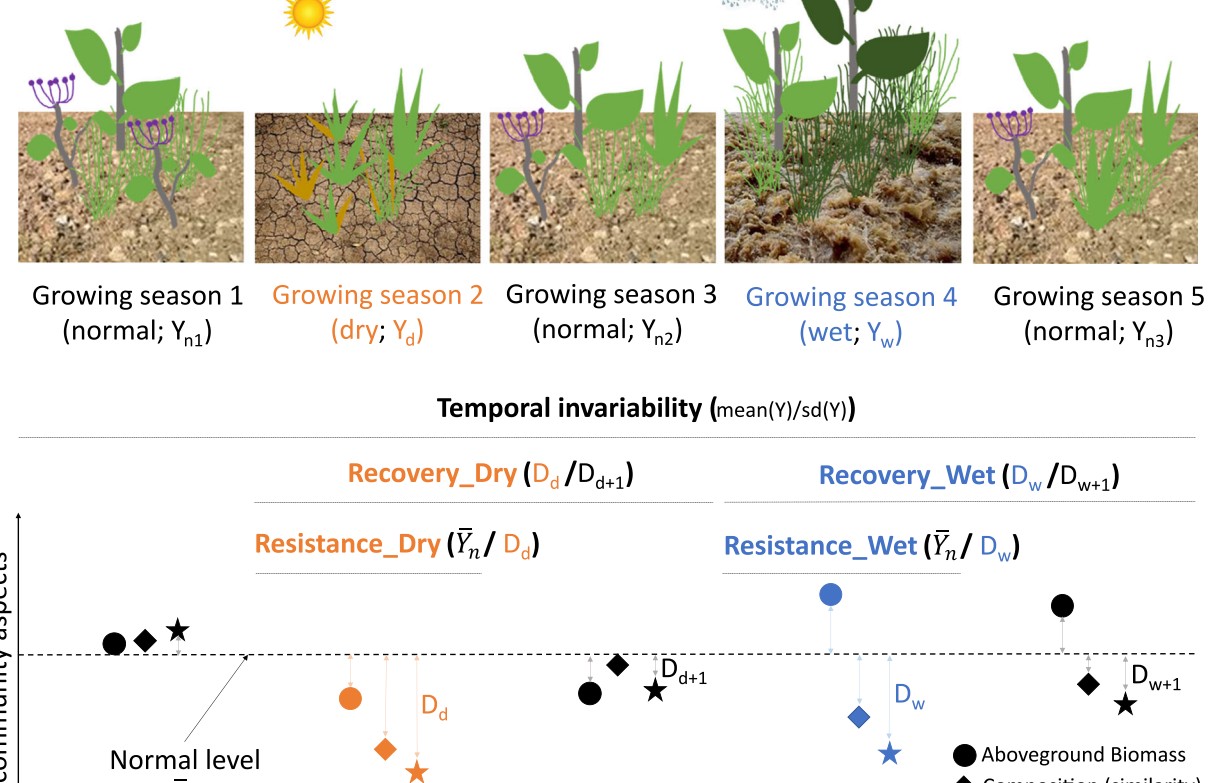

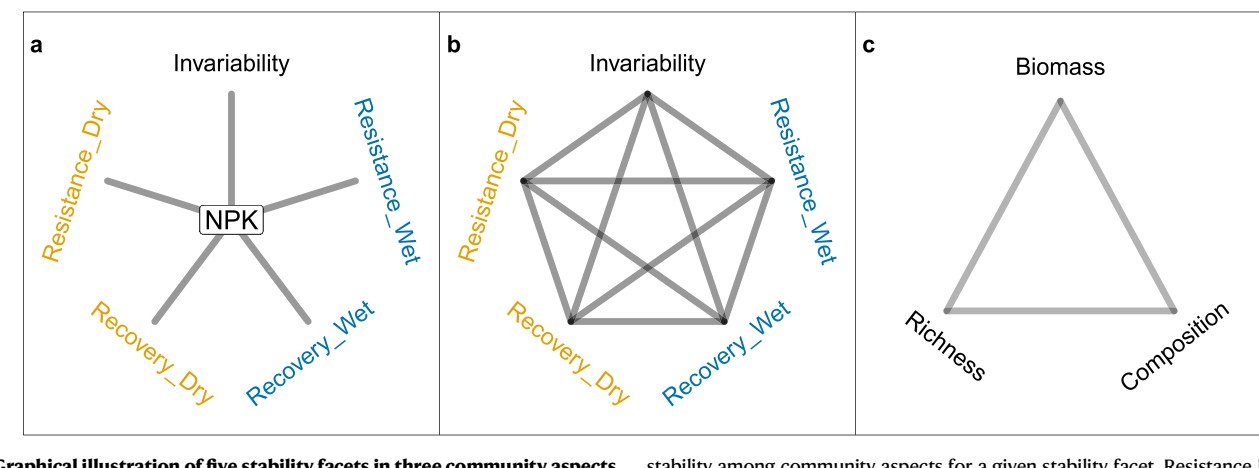

**Fig. 1 | Graphical illustration of five stability facets in three community aspects investigated in this study.** Methods used for quantifying stability facets are shown. We investigate the effects of nutrient addition (NPK) on (**a**) each of the five stability facets within each community aspect, (**b**) pairwise correlations among stability facets within each community aspect, and (**c**) pairwise correlations of stability among community aspects for a given stability facet. Resistance_Dry: resistance during dry growing seasons; Resistence_Wet: resistance during wet growing seasons; Recovery_Dry: recovery after dry growing seasons; Recovery_Wet: recovery after wet growing seasons.

for regulating ecosystem functions and maintaining ecosystem stability. For instance, higher invariability in community composition usually leads to higher invariability in aboveground biomass in grasslands[17–19]. But low stability in community composition (large compositional variation) can also be associated with high biomass stability if different species exhibit compensatory dynamics over time[20,21]. Higher species richness (often the mean) has been shown to enhance temporal invariability of aboveground biomass because population decreases in some species due to climate extremes may be compensated by increases in other species that can endure harsh climate conditions[22]. This relationship can be modulated by global change factors such as eutrophication and aridity[23,24]. Moreover, species richness itself is likely to vary over time under climate change, and the correlations among stability in species richness and that in aboveground biomass or community composition are largely unclear. A simultaneous understanding of the various facets of stability in multiple community aspects is crucial for predicting and managing ecosystem stability in the face of global environmental change such as eutrophication.

Nutrients and water availability are essential for the survival and growth of plants[25], but imbalance in their availability can have negative impacts on various community aspects[26]. Numerous studies have investigated the effects of eutrophication on the temporal invariability of aboveground biomass, its resistance during, and recovery after dry climate extremes, but found mixed results (with positive, negative, or no effects of eutrophication on these stability facets all reported)[27–35]. In comparison, only a few studies have explored the impact of eutrophication on resistance of biomass during and recovery after wet climate extremes[32]. Furthermore, eutrophication has been well documented to decrease species richness and cause vegetation shifts toward domination by fast-growing (typically associated with high leaf nutrients) and invasive species[36,37]. However, whether and how eutrophication affects the stability of species richness and composition remain an open question.

Additionally, eutrophication may alter the effective dimensionality of stability by changing the correlations among stability facets. On the one hand, eutrophication can enhance interspecific competition and deterministic community assembly processes, thus strengthening the correlation among different stability facets[11,12]. But on the other hand, eutrophication may promote stochastic community assembly via increasing soil fertility and productivity, possibly weakening the correlations among different stability facets[38]. Indeed, different facets of stability measured in different community aspects may respond differentially (in both direction and magnitude) to eutrophication[18,32], leading to either weakened or strengthened correlations among

stability measures. A recent study in a grassland finds that eutrophication does not alter relationships among stability in community biomass and composition[18]. Overall, a systematic investigation of the correlations among stability facets in various community aspects and their responses to eutrophication is still lacking.

Using 55 grassland sites spanning 5 continents with at least 4 years of standardized experimental nutrient addition, we tested whether nutrient addition alters five facets of stability, measured for three focal plant community aspects, and correlations among them (Fig. 1). The five stability facets are temporal invariability, resistance during dry and wet growing seasons, and recovery after dry and wet growing seasons. The three plant community aspects are aboveground biomass, community composition, and species richness. To enable comparison among sites with varying conditions, we quantified resistance as the inverse of the proportional deviation of a community aspect from its normal levels during a dry or wet growing season. We quantified recovery as the ratio of deviation in a community aspect during to that after a dry or wet growing season following ref. 39. We categorized dry, normal, and wet growing seasons based on each site's historical standardized precipitation–evapotranspiration index (SPEI; dry: ≤25th percentile; wet: ≥75th percentile; normal: 25–75th percentile of SPEI; Supplementary Figs. 1–5; Supplementary Table 1–2; see "Methods") following ref. 39. In total, we recorded 150 dry, 247 normal, and 131 wet growing seasons across all sites during the study period (see Supplementary Fig. 2 for the number of dry and wet growing seasons at individual sites). We find that nutrient addition reduces the temporal invariability and resistance of species richness and community composition during dry and wet growing seasons, but it does not affect those of biomass. Our analyses also reveal high dimensionality of grassland stability, i.e., low correlations among different stability measures, under both ambient and eutrophic conditions.

## Results and discussion
### Effects of nutrient addition on different stability measures
Nutrient addition reduced temporal invariability and resistance of community composition and species richness during dry and wet growing seasons, but it did not affect any stability facets of biomass investigated (Fig. 2; Supplementary Table 3; see Supplementary Fig. 6 for stability facets at individual sites). Across our study sites, nutrient addition increased aboveground biomass during normal growing seasons and deviation (from the normal levels) during dry and wet growing seasons (Supplementary Notes; Supplementary Fig. 7; Supplementary Table 4). But the proportional deviation was similar under ambient and nutrient addition conditions (Supplementary Notes; Supplementary Fig. 7), resulting in no discernable impact of nutrient

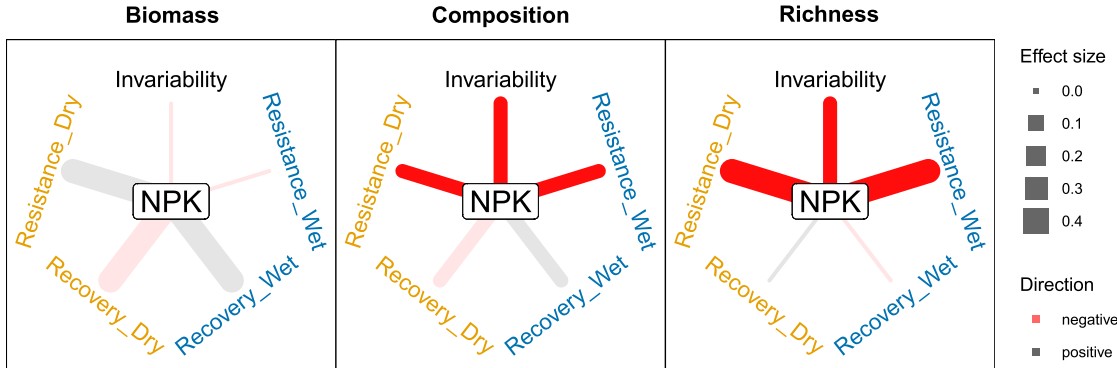

**Fig. 2 | Effects of nutrient addition (NPK) on each of the five stability facets in each of the three community aspects.** Resistance_Dry: resistance during dry growing seasons; Resistence_Wet: resistance during wet growing seasons; Recovery_Dry: recovery after dry growing seasons; Recovery_Wet: recovery after wet growing seasons. Saturated line colors represent significant treatment effects at $p \leq 0.05$ and faded line colors represent non-significant treatment effects. The significance of treatment effects was assessed using $t$ test. See Supplementary Table 3 for test statistics, effect sizes, standard errors of the effect size, degrees of freedom, and $p$ values for the two-tailed test.

addition on resistance and invariability. Two processes may underlie the robustness of biomass responses to dry and wet growing seasons under nutrient addition, as compared to species richness and community composition. First, species turnover may compensate for biomass loss resulting from species loss[40]. That is, when local communities experience species loss, the vacant niches or spaces can be quickly filled by the remaining or newly colonized species, thus maintaining total biomass at a similar level[40,41]. Second, biomass changes are more driven by dominant species, which may be less sensitive to climate extremes under nutrient addition[32] (see the next paragraph). Our findings indicate that conserving plant diversity and community composition (e.g. pollinator plants or endangered species) may be more challenging than maintaining biomass production (e.g. agricultural grasslands) during climate extremes under eutrophication.

Extending our current understanding, we showed that nutrient addition reduced resistance of species richness and community composition during both dry and wet growing seasons. For community composition, nutrient addition reduced its resistance during dry and wet growing seasons by altering species abundance distributions during normal growing seasons and increasing compositional changes relative to the normal levels (Supplementary Notes; Supplementary Fig. 8; Supplementary Table 4). For species richness, nutrient addition decreased its resistance during both dry and wet growing seasons but through different processes. Specifically, nutrient addition reduced the resistance of species richness during dry growing seasons by the combined effects of reducing the average of normal levels and increasing deviation from the normal levels, whereas it reduced the resistance during wet growing seasons primarily by reducing the normal levels (Supplementary Notes; Supplementary Fig. 9; Supplementary Table 4). To determine whether nutrient addition modulated resistance of species richness mainly by influencing rare species, we calculated species diversity weighted by species cover. We found that nutrient addition similarly reduced invariability and resistance of cover-weighted species diversity (e.g., Hill number equals 0, 1, and 2) during dry growing seasons. But the effects of nutrient addition on

resistance during wet growing seasons decreased with increasingly high weights for abundant species (non-significant for Hill number equal to 2; Supplementary Fig. 10). This suggests that dominant plant species may be more resistant than rarer species during wet growing seasons under eutrophication.

Nutrient addition did not impact recovery after dry or wet growing seasons for any stability measures investigated compared with ambient conditions (Fig. 2; Supplementary Table 3). But the absolute deviations were much greater under nutrient addition than those under ambient conditions especially for community composition and biomass, during and one year after dry and wet growing seasons (Supplementary Notes; Supplementary Figs. 7–9; Supplementary Table 4). Such information on absolute deviations may complement the results based on relative deviations to guide conservation and management for different purposes (e.g., agricultural yields) at specific sites.

## Effects of nutrient addition on the dimensionality of stability

We first analyzed the correlation among stability facets within each community aspect and tested how they responded to nutrient addition. We calculated pairwise Pearson correlation coefficients among stability facets (10 pairs from 5 facets) for each community aspect in either treatment within each site. Under ambient conditions, some pairs of stability facets were significantly correlated in biomass (4/10 pairs), community composition (4/10), and species richness (2/10), respectively. Overall, this suggests a relatively high dimensionality of stability (Fig. 3; Supplementary Table 5), in line with previous studies showing that the stability of ecosystems cannot be characterized by only one or two facets[8,10], and not a single strategy can promote all facets of stability simultaneously. These correlations or lack thereof were generally robust under nutrient addition, although certain pairs of stability facets were weakened or enhanced (Fig. 3; Supplementary Table 5). For instance, nutrient addition resulted in a negative correlation between temporal invariability and recovery after wet growing seasons in community composition, which were not correlated under ambient conditions.

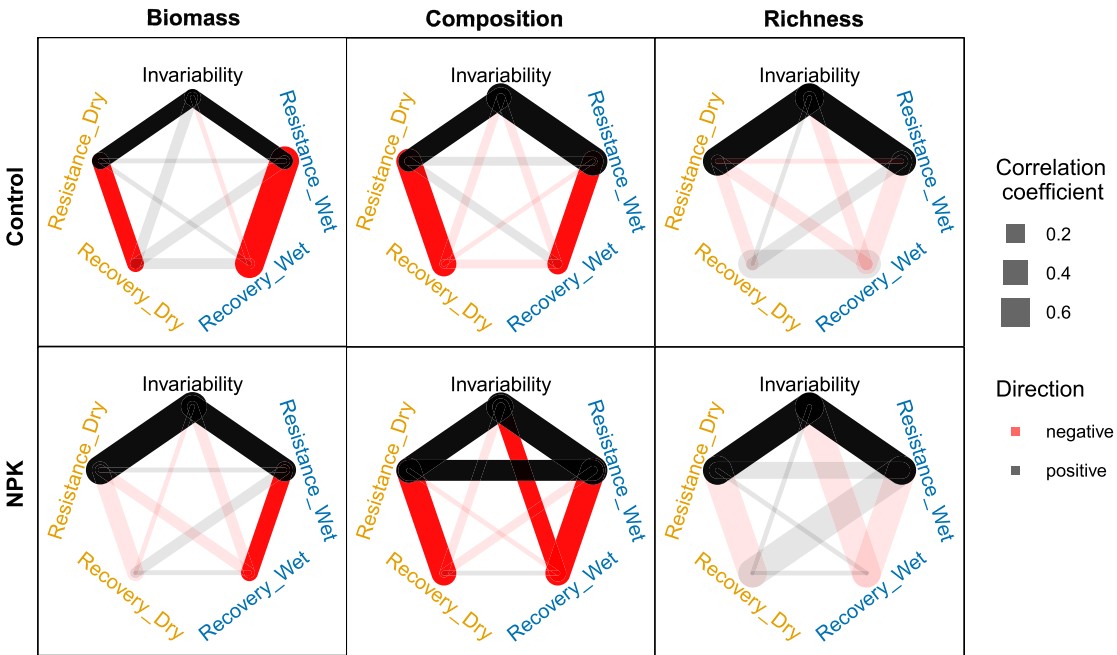

**Fig. 3 | Pairwise correlations among the five stability facets in each of the three community aspects under ambient (control) and nutrient addition (NPK) conditions.** Resistance_Dry: resistance during dry growing seasons; Resistence_Wet: resistance during wet growing seasons; Recovery_Dry: recovery after dry growing seasons; Recovery_Wet: recovery after wet growing seasons. Saturated line colors represent significant correlations, corresponding to 95% confidence intervals of the correlation coefficients that do not overlap with 0. Faded line colors represent non-significant correlations. See Supplementary Table 5 for test statistics and 95% confidence intervals for each correlation coefficient.

Notably, temporal invariability was positively correlated with resistance, but not recovery, under both treatments for all three community aspects investigated (Fig. 3; see Supplementary Figs. 11–13 for correlations at individual sites). Our result is in line with the finding from manipulated biodiversity experiments that infers that grassland species richness increases temporal invariability by enhancing resistance rather than recovery[39]. Combined, these results indicate that in spite of the expected positive associations between temporal invariability with both resistance and recovery by definition and quantification, long-term temporal invariability of different community aspects relies on their short-term responses during climate extremes rather than their subsequent recovery. Thus, strategies aimed at enhancing community resistance, such as using drought and flood resistant genotypes or species, are also crucial for promoting the temporal invariability of ecosystem properties in eutrophic conditions.

For both aboveground biomass and community composition, resistance during and recovery after dry or wet growing seasons were generally negatively correlated under both ambient and nutrient addition conditions (Fig. 3; Supplementary Table 5). This indicates that biomass and community composition that were more impacted during dry and wet growing seasons (relative to normal levels) also recovered faster, regardless of nutrient conditions (changes were reversible). Such a trade-off between resistance and recovery is important for the maintenance of community composition and functions in the face of climate extremes[42]. In comparison, such a trade-off was not observed for species richness, implying that species richness may be more difficult to recover or its recovery may take more time after perturbations.

Next, we tested the correlations among stability of community aspects for a given stability facet and their responses to nutrient addition. We found that, for each stability facet, the correlations among the stability of the three community aspects were generally weak, under both ambient and nutrient additions (Fig. 4; see Supplementary Fig. 14 for correlations at individual sites; Supplementary Table 6). The consistently weak correlations among stability of aboveground biomass, species richness, and community composition may indicate differential responses of different community aspects to climatic fluctuations. The low correlation between stability of community composition and species richness may also suggest that community compositional changes were mainly driven by species replacement but not species loss[17]. Our findings differ from results from a recent meta-analysis showing that compositional stability and biomass stability are positively correlated[10]. This discrepancy may be

understood from the different definitions of stability facets and different ways of quantification for correlations. For instance, while our study calculated correlations among stability measures within experimental sites, the meta-analysis derived correlations across sites along environmental gradients which might mediate positive correlations between stability measures. Our results suggest that stability of species richness, community composition, and biomass represent separate dimensions. Thus, different and context-dependent strategies may be required for effective management for different community aspects.

## Robustness and limitations

To address the robustness of our results, we re-performed the analyses (i) using more extreme thresholds of SPEI to define dry and wet growing seasons (dry: ≤10th percentile; wet: ≥90th percentile; Supplementary Figs. 15–17); (ii) after removing the long-term linear trends of SPEI (Supplementary Figs. 18–20); and (iii) based on 22 sites with at least 10-year nutrient addition and observations (Supplementary Figs. 21–23). These analyses led to similar patterns as those presented above: (i) nutrient addition decreased the temporal invariability and resistance during dry and wet growing seasons for composition and richness, but not for biomass; (ii) the majority of pairs of stability measures were uncorrelated, while nutrient addition changed the correlations for certain pairs of stability measures. That said, the specific pairs of stability measures that exhibit significant correlations do change due to differences in the number and identity of sites included in these additional analyses. This was particularly the case for correlations among stability measures after removing long-term linear trends of SPEI. One limitation of our analyses is to appropriately quantify resistance and recover in the presence of consecutive climate extremes (e.g. dry and dry, dry and wet, or wet and wet growing seasons). Existing theories mostly consider a single pulse perturbation to quantify resistance and recovery (e.g., Fig. 1), making it challenging to disentangle ecosystem responses to consecutive extreme growing seasons. In such cases, a previous extreme growing season may affect the community's response to the subsequent extreme growing seasons, i.e., legacy effects[43]. New theory and methodology are needed to address ecosystem responses to repeated and consecutive perturbations.

## Implications

Our study represents a globally coordinated efforts and provides a comprehensive test of the effects of nutrient addition on different

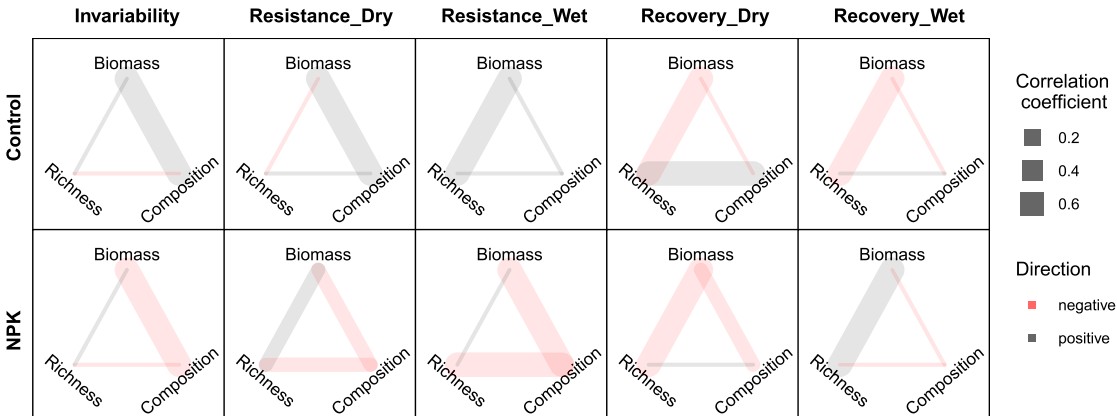

**Fig. 4 | Pairwise correlations among stability of the three community aspects for a given stability facet under ambient (control) and nutrient addition (NPK) conditions.** Resistance_Dry: resistance during dry growing seasons; Resistence_Wet: resistance during wet growing seasons; Recovery_Dry: recovery after dry growing seasons; Recovery_Wet: recovery after wet growing seasons. Saturated line colors represent significant correlations, corresponding to 95% confidence intervals of the correlation coefficients that do not overlap with 0. Faded line colors represent non-significant correlations. See Supplementary Table 6 for test statistics and 95% confidence intervals for each correlation coefficient.

stability facets measured for multiple community aspects and their correlations. Previous studies usually focus on the effects of nutrient addition on the mean, and to a lesser extent, temporal trends in species richness, and/or overall change in community composition, which in turn are used to predict temporal invariability of biomass[17,44,45]. While these studies do not take into account the stability of species richness and community composition, our study shows that nutrient addition decreases their short-term resistance during dry and wet growing seasons and their long-term temporal invariability. Overall, our analyses demonstrate high dimensionality of grassland stability, that is, consistently low correlations among stability facets within and among community aspects under ambient and eutrophic conditions. Our results suggest that different stability measures may be regulated by independent mechanisms and processes, underscoring the need to consider the multidimensional responses of ecosystems to environmental changes. By disentangle the concurrent impacts of global changes on ecosystems, our results provide new insights and opportunities to achieve a more holistic understanding of grassland stability in a changing world.

## Methods

### Experimental design

The study sites are part of the NutNet experiment[46,47] (Supplementary Fig. 1 and Supplementary Table 1). Plots of 5 m × 5 m were assigned to ten treatments in a randomized block design, typically with three blocks per site. For the analyses here, we select plots assigned to one of two treatments: Control and Fertilized by NPK$_{+\mu}$. NPK$_{+\mu}$ treatment plots were fertilized with nitrogen (N), phosphorus (P), and potassium (K) with a combination of micronutrients and macronutrients (Fe, S, Mg, Mn, Cu, Zn, B, and Mo) as a one-time addition to the potassium treatment (K$_{+\mu}$). The micronutrient mix was applied once at the start of the experiment at a rate of 100 g m$^{-2}$. N was supplied as time-release urea ((NH$_2$)$_2$CO), P was supplied as triple superphosphate (Ca(H$_2$PO$_4$)$_2$), and K as potassium sulfate (K$_2$SO$_4$). N, P, and K were added annually at rates of 10 g m$^{-2}$ y$^{-1}$. More details in experimental design can be found in ref. 46.

Data was retrieved in November 2022. The 55 sites included in this study met the following criteria: (1) plots were arranged in 3 blocks; (2) ≥ 4 years of post-treatment measurement; (3) during experimental years, at least one dry or wet growing season was recorded (see "Defining climate extremes" for more details). These sites span five continents and include a wide range of grassland types. See Supplementary Figs. 1–2, and Supplementary Table 1 for details of sites selected, experimental years, geolocation, growing season, and grassland types.

### Sampling protocol

All NutNet sites followed standard sampling protocols. A 1 × 1 m subplot within each plot was permanently marked for annual measurement of plant community composition. Species cover (%) was estimated visually for all species in the subplots; the total cover of living plants can exceed 100 % for multilayer canopies. Aboveground biomass was measured within the treatment plot, adjacent to the permanent subplot, by clipping all aboveground biomass within two 1 × 0.1 m strips (in total 0.2 m$^2$), which were moved each year to avoid resampling the same location. For shrubs and subshrubs occurring in strips, we collected all leaves and current year's woody growth. Biomass was dried at 60 °C (to constant mass) before weighing to the nearest 0.01 g, and expressed as g m$^{-2}$. At most sites, cover was recorded once per year at peak biomass before fertilization. At some sites with strong seasonality, cover was recorded twice per year to include a complete list of species. For those sites, the maximum cover for each species and total biomass were used in the following analyses. The taxonomy was adjusted within sites to ensure consistent naming over time. Specifically, when individuals could not be identified as

species, they were aggregated at the genus level but referred to as "species" for simplicity.

### Defining climate extremes and the five stability facets for the three community aspects

We used the standardized precipitation–evapotranspiration index (SPEI) to classify climate events for each site[39]. SPEI was calculated as the standardized (z-score) water balance over the growing season each year (sum of precipitation – sum of evapotranspiration; mm) from 1901 to 2021. Precipitation and potential evapotranspiration used to calculate SPEI were downloaded from https://crudata.uea.ac.uk/cru/data/hrg/cru_ts_4.06[48]. Potential Evapotranspiration is calculated using the Penman-Monteith formula taking into account the rate of change of saturation specific humidity with air temperature, net irradiance, ground heat flux, air temperature at 2 meters, wind speed at 2 meters, and vapor pressure deficit[49]. At each site, we then classified the growing seasons into dry, normal, and wet using the cutoff of 0.67 or 1.28 times of standard deviation (SD). A cutoff of 0.67 SD corresponds to a definition of dry or wet growing season occurring once every four years (i.e., dry: ≤25th percentile; wet: ≥75th percentile), and that of 1.28 SD corresponds to once per decade (i.e., dry: ≤10th percentile; wet: ≥90th percentile). In both cases, normal growing seasons were defined as −0.67 sd < SPEI < 0.67 sd. A cutoff of 0.67 SD resulted in a much higher number of dry and wet growing seasons occurring at the 55 sites (Supplementary Fig. 2), which increases the power of our statistical analyses. Results were similar when defining dry and wet growing seasons using the cutoff of 1.28 SD (resulting in 66 dry, 247 normal, and 58 wet growing seasons in 44 sites). Therefore, we present the results based on the cutoff of 0.67 SD in the main text and those based on the cutoff of 1.28 SD in the supplementary (Supplementary Figs. 15–17). To reduce confounding effects of dry and wet growing seasons occurring consecutively on the calculation of resistance and recovery, we selected experimental years using the following two criteria. First, if two consecutive extreme growing seasons were of different kinds (e.g., dry followed by wet), the former growing season was ignored for the calculation of recovery, and the later growing season was ignored for the calculation of both resistance and recovery. Second, when two (or more) extreme growing seasons of the same kind happen consecutively (e.g., wet followed by wet), recovery was only calculated for the later growing season, which must be followed by a normal growing season (or a same kind but less extreme growing season when using 1.28 SD as the cutoff). See Supplementary Table 2 for all combinations of three consecutive growing seasons and selection of the growing seasons for calculating resistance and recovery. We illustrated by example the selection of the years and calculation of resistance and recovery at three blocks at site Look.us (Supplementary Fig. 4). Also, we showed years used for resistance and recovery at individual sites (Supplementary Fig. 5). Furthermore, we summarized average changes and absolute deviations in aboveground biomass, species richness, and community composition (from their normal levels) during and one year after dry and wet growing seasons in both treatments (Supplementary Figs. 7–9). To facilitate an intuitive understanding of each stability facets and the correlations among them, we present the five stability facets in the three community aspects at each site in Supplementary Fig. 6 and pairwise correlations among stability measures in Supplementary Figs. 11–14.

Methods used for quantifying stability facets are illustrated in Fig. 1. Specifically, temporal invariability in aboveground biomass and species richness was calculated as $\frac{\mu}{\sigma}$, where μ is the mean in aboveground biomass or species richness over the experimental years. $\sigma$ is the standard deviation of aboveground biomass or species richness over time, which was calculated after detrending to remove variation due to directional change over time. That is, we first used linear regression (function "lm") to fit aboveground biomass or species richness against experimental years for each subplot, we then used the

residuals from this model to calculate the standard deviation. Following ref. [39], resistance and recovery were calculated as $\frac{\bar{Y}_n}{|\bar{Y}_e - \bar{Y}_n|}$ and $\frac{|Y_e - \bar{Y}_n|}{|Y_{e+1} - \bar{Y}_n|}$, where $\bar{Y}_n$, $Y_e$, and $Y_{e+1}$ are aboveground biomass or species richness during normal growing seasons (average over all normal growing seasons), during an extreme, and one year after an extreme growing season. Stability facets in biomass and species richness were log-transformed to improve homogeneity of variance. The composition-related facets of stability were calculated using Bray–Curtis dissimilarity metric based on cover data[50,51]. Values of Bray-Curtis dissimilarity ranges from 0 to 1, with higher dissimilarity being closer to 1. We used similarity (i.e., 1 - dissimilarity) to measure stability in community composition, with higher values indicating higher stability. Temporal invariability was calculated as the overall community similarity over all experimental years using the function "beta.multi.abund" from the R package "betapart"(version 1.6)[50]. The average cover for all species during normal growing seasons under each treatment within a block was constructed as a reference community for calculating resistance and recovery. Resistance was calculated as the similarity of the plant community under an extreme growing season compared with the reference (values ranged from 0 to 1) using the R function "beta.pair.abund". Similarly, we calculated similarity of the plant community one year after an extreme growing season compared with the reference. Recovery was then calculated as the ratio of similarity of the community one year after to that during an extreme growing season (ranged from 0 to inf). Resistance and recovery for all three community aspects were averaged over years to match the data structure of temporal invariability. For all stability facets in the three community aspects, higher values represent higher stability.

## Statistical analysis

All analyses were performed in R (version 4. 2.0)[52]. We used linear mixed-effects models (function "lme") from the R package "nlme" (version 3.1.157) for the following analyses[53]. First, we tested whether nutrient addition impacted each stability facet for each community aspect. In these models, treatment (control vs. nutrient addition) was the fixed effect, and site and block nested within site were random effects. To assess whether rare species were more sensitive than common and dominant species during and after dry and wet growing seasons under nutrient addition, we calculated effective species diversity corresponding to Hill numbers Q ranging from 0 to 2 (an increase in Q indicating greater weights of abundant species)[54]. Second, we quantified the effects of nutrient addition on relationships among the five stability facets for each community aspect. We calculated Pearson correlation coefficients between every pair of stability facets for each treatment within each site, and then used the function "lme" to test the effect of treatment (fixed effect) on these correlation coefficients, with site as the random effect. Third, we examined the effects of nutrient addition on relationships among stability of the three community aspects (for each stability facet). Similarly, we calculated Pearson correlation coefficients for each stability facet among all pairs of community aspects for each treatment at each site, and then used the function "lme" to test the effect of treatment (as fixed effects) on these correlation coefficients, with site as the random effect. We tested the robustness of our results in the following three ways. First, we examined more extreme thresholds in defining dry and wet growing seasons, as mentioned above. Second, we tested the temporally linear trend of SPEI for each site and re-analyzed the data after removing the trends of SPEI, because a positive (negative) trend of SPEI would lead to higher likelihood to detect extreme wet (dry) years at the end of the time series. Third, as the experiment duration of our study sites ranged from 4 to 15 years, we re-performed the above analyses using a subset of 22 sites that have run experiments for at least 10 years.

## Reporting summary

Further information on research design is available in the Nature Portfolio Reporting Summary linked to this article.

## Data availability

The NutNet data are publicly available on the Environmental Data Initiative (EDI) (https://doi.org/10.6073/pasta/583874460a0af70f93d3eee2f22f9a13). The climate data are available on https://crudata.uea.ac.uk/cru/data/hrg/cru_ts_4.06. The raw data used and processed data generated in this study have been deposited in the Figshare (https://doi.org/10.6084/m9.figshare.22639399).

## Code availability

The R codes used to produce results in this study have been deposited in the GitHub (https://github.com/chqq365/multidimensional-responses-of-stability.git.) and archived through Zenodo (https://doi.org/10.5281/zenodo.8292710).

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

## Acknowledgements

We thank researchers from the NutNet who contributed data to our analysis but are not listed as authors (supplementary Table 7). We thank the Minnesota Supercomputing Institute for hosting project data and the Institute on the Environment for hosting Network meetings. Nitrogen fertilizer was donated to NutNet by Crop Production Services, Loveland, CO. This experiment is funded by individual researchers at the site scale. To Companhia das Lezirias for providing access to the study site.

## Author contributions

Q.C., S.W., and Y.H. conceived the study; Q.C., S.W., and Y.H. developed the methodology and analyzed data with contribution from J.D.B.; J.D.B. and Y.N. checked the R codes; E.T. B., J.D.B., E.W.S., W.S.H., N.E., Y.L., Y.M.B., J.A.C., C.R., I.D., S.A.P., P.D., A.E., J.M.H.K., J.P.M., A.E., J.W.M., A.C.R., M.C.C., M.N.B, R.V., I.C.B., Y.N., A.J., C.J. S., D.S.G., A.S.M., J.A., and Y.H. contributed data. Q.C. visualized the results; Q.C., S.W., and Y.H. wrote the original draft of the manuscript; other authors (E.T. B., J.D.B., E.W.S., W.S.H., N.E., Y.L., Y.M.B., J.A.C., C.R., I.D., S.A.P., P.D., A.E., J.M.H.K., J.P.M., A.E., J.W.M., A.C.R., M.C.C., M.N.B, R.V., I.C.B., Y.N., A.J., C.J. S., D.S.G., A.S.M., J.A.) reviewed and edited the manuscript. E.T.B. and E.W.S. are Nutrient Network coordinators. Please see Supplementary Table 8 for more details.

## Funding

National Natural Science Foundation of China grant 31988102, 32122053 (SW). National Science Foundation grant NSF-DEB-1042132 (ETB, EWS; for NutNet coordination and data management) National Science Foundation grant NSF-DEB-1234162 (ETB, EWS; for Long-Term Ecological Research at Cedar Creek). National Science Foundation grant NSF-DEB-1831944 (ETB, EWS; for Long-Term Ecological Research at Cedar Creek). Irish Research Council Laureate Awards 2017/2018 IRCLA/2017/60 (YMB). The Portuguese Science Foundation (FCT) for Forest Research Center funding (UIDB/00239/2020).

## Competing interests

The authors declare no competing interests.

## Additional information

[1]Institute of Ecology, College of Urban and Environmental Sciences, Peking University, Beijing, China. [2]German Centre for Integrative Biodiversity Research (iDiv), Puschstrasse 4, 04103 Leipzig, Germany. [3]Department of Ecology, Evolution, and Behavior, University of Minnesota, St. Paul, MN, USA. [4]School of Environmental and Forest Sciences, University of Washington, Seattle, WA, USA. [5]Department of Physiological Diversity, Helmholtz Center for Environmental Research–UFZ, Permoserstrasse 15, 04318 Leipzig, Germany. [6]Martin Luther University Halle-Wittenberg, am Kirchtor 1, 06108 Halle (Saale), Germany. [7]Institute of Biology, Leipzig University, Leipzig, Germany. [8]MPG Ranch and University of Montana, Missoula, MT, USA. [9]School of Natural Sciences, Zoology, Trinity College Dublin, Dublin, Ireland. [10] Department of Geography, King's College London, 30 Aldwych, London WC2B 4BG, UK. [11]Hawkesbury Institute for the Environment, Western Sydney University, Locked Bag 1797, Penrith, NSW 2751, Australia. [12]Instituto de Investigaciones Marinas y Costeras (IIMyC), FCEyN, UNMdP-CONICET, CC 1260 Correo Central, B7600WAG Mar del Plata, Argentina. [13]Institute of Ecology and Evolution, University Jena, Jena, Germany. [14]Health & Environmental Sciences, Xián Jiaotong Liverpool University, Suzhou, China. [15]Department of Biology, Texas State University, San Marcos, TX 78666, USA. [16]Ecology and Genetics, University of Oulu, Oulu, Finland. [17]Department of Environment and Genetics, La Trobe University, Bundoora 3086 VIC, Australia. [18]Swiss Federal Institute for Forest, Snow and Landscape Research WSL, Zuercherstrasse 111, 8903 Birmensdorf, Switzerland. [19]Forest Research Centre, Associate Laboratory TERRA, School of Agriculture, University of Lisbon, Lisbon, Portugal. [20]Centre for Applied Ecology "Prof. Baeta Neves" (CEABN-InBIO), School of Agriculture, University of Lisbon, Lisbon, Portugal. [21]Faculty of Environmental and Forest Sciences, Agricultural University of Iceland, Hvanneyri, Iceland. [22]Disturbance Ecology and Vegetation Dynamics, Bayreuth Center of Ecology and Environmental Research (BayCEER), University of Bayreuth, Bayreuth, Germany. [23]Lancaster Environment Centre, Lancaster University, Lancaster LA1 4YQ, UK. [24]Department of Entomology, University of Maryland, College Park, MD, USA. [25]Department of Integrative Biology, University of Guelph, Guelph, ON, Canada. [26]Ecology and Biodiversity Group, Department of Biology, Utrecht University, Padualaan 8, 3584 CH Utrecht, The Netherlands. ✉e-mail: shaopeng.wang@pku.edu.cn

