## [Peer review file · Nature Communications]

REVIEWER COMMENTS

Reviewer #1 (Remarks to the Author):

In this manuscript the authors examine a very impressive dataset of 55 grassland sites that are part of the Nutrient Network to see how nutrient additions influence several facets of stability. They find that nutrient additions tend to decrease temporal invariability and resistance to wet and dry extremes for species composition and richness, but not biomass. They also find that invariability is positively related to resistance across nutrient treatments and community aspects. These results are interesting and they provide useful information about how global changes interact to affect grassland plant community stability from several perspectives. The manuscript is well written, but I have several overarching concerns and comments that I believe would improve the manuscript.

First, I think the manuscript would benefit greatly from showing some of the data directly, rather than simply summarizing correlations (Figs 2-4). I don't even think I saw any of the invariability, resistance, or recovery values reported in the supplementary materials, so it is very hard to get an idea of what the data look like. Specifically, it's not clear how invariable places are, whether they recover, etc. I understand that this could be a bit messy, and might need to be put in the supplement, but finding a way to show the values used in the correlations would improve the ability to interpret your results.

Second, the results and discussion section feels mostly like a results section, with only very shallow discussion included for your results. I think your manuscript could benefit greatly from more detailed interpretation of your results by putting the results into context and elaborating on some of the statements that you provide. For example, you mention a few times that we should focus on increasing resistance rather than recovery in ecosystems. Can you provide any information about how you might do that?

Third, I think you need stronger justification for focusing on wet and dry years. Could you also focus on hot years? Including more detail up front about why wet and dry years are especially interesting/important would be good.

Fourth, some of your sites only have 4 years of data and it made me wonder whether that is enough to calculate stability? Many of the calculations rely on calculating a "baseline" or normal level, which seems like it would become increasingly sensitive to random variation when you have few samples. It doesn't look like many sites have only four years, but it might be worth checking how sensitive your results are to time series length.

Finally, the main message of your manuscript seems to be that invariability and resistance are related, and not related to recovery, which seems like it should be that way based on how the metrics are calculated (higher resistance is less deviation from the mean, which is also essentially invariability, and where there is recovery, there cannot be invariability and resistance must be relatively low). So, I think reporting those results are not especially profound. I think it might be more interesting to focus on where these expected relationships are not so obvious and why that might be.

In addition, here are a few line by line comments:

Abstract:

Line 50-51: The phrasing here makes it hard to pick out the 5 dimensions. Maybe make it a numbered list in the parentheses?

Line 55-56: Is this not somewhat mathematically expected, as resistance should be related to invariability because both are a lack of change through time?

Line 57-58: I'm not sure why this follows from the previous sentence. Are you trying to say that the

grasslands that you studied did not show recovery after dry and wet years? It still seems like recovery should be an important element of stability.

Introduction:

Lines 65-67: Citation for this statement?

Line 67: Diversity and functionality of what?

Line 91-93: Not sure what you mean by "resp. negatively", as you later say that if facets are negatively correlated, then you can't optimize both.

Lines 100-103: I think providing more justification for picking these three community aspects would be useful, especially to set up expectations about which aspects should be more or less stable and which should be correlated and how.

Results and discussion:

No specific comments, see overarching comments above.

Methods:

Lines 279-280: Is it potentially problematic if your sites have directional trends in SPEI from 1901-2021? It seems like it may be a problem because if you have a positive trend through time, you will be more likely to detect extreme wet years at the end of the time series. Did you check this by detrending the SPEI data?

There is not enough information about how the resistance and recovery metrics were calculated. For example, I don't understand why invariability and resistance should range between -1 and 0 and not 0 to 1. It also seems theoretically possible that resistance could be greater than 1 (see aboveground biomass for a wet year in figure 1). I may be misunderstanding something, but I think this needs more clarity.

Figures:

Figure 1: This is very helpful for visualizing the stability facets. Nice job.

Reviewer #2 (Remarks to the Author):

SUMMARY

The authors have conducted an interesting study in which they investigate how different facets of ecological stability, the resistance during and the recovery after extreme climate events of plant communities are affected by nutrient addition in grasslands. Using data collected in 55 grassland sites during at least 4 years of nutrient addition experiments (NutNet experiments), they test how nutrient addition affects the correlation between the stability, the resistance during and the recovery after dry and wet growing seasons of three plant community aspects: aboveground biomass, the composition of the plant community and its species richness. They show that nutrient addition reduces the stability and resistance of species richness and community composition during dry and wet climate events. However, the stability and resistance of aboveground biomass was not affected by nutrient addition. The stability and the resistance of the different community aspects were positively correlated - independently of nutrient addition - but it was not correlated with recovery.

Overall, I think that these results provide useful information about the consequences of two important global change drivers for ecosystem functioning: nutrient addition and climate changes. I have only few minor concerns with that manuscript:

1. I was wondering why the authors used the term 'temporal invariability' and not the term 'stability'? I think that the term 'stability' is more often used and understandable by a wide readership. I would therefore consider to homogenizing the wording and using this term instead of 'temporal invariability'.
2. The references cited are not always adapted to support the ideas developed, especially in the 'Introduction' section (see specific suggestions below). I think that the authors should check carefully that they are using appropriate references.
3. I was left a bit disappointed with the 'Results and discussion' section. I am aware that the authors are limited by the word count but I think that the discussion points could be expanded. For now, the discussion is quite shallow (see also specific suggestions).
4. Finally, in this study, the authors look at how nutrient addition affect the relationship between different stability facets, including the resistance during and recovery after a dry or wet event. I think it is also important to keep in mind and acknowledge that climatic conditions can modulate the relationships between biodiversity and ecosystem functioning (see e.g. Shi et al. 2016, Ma et al. 2017, Garcia-Palacios et al. 2018).

Below, I present more specific suggestions that may help, I hope, to improve the manuscript.

SPECIFIC SUGGESTIONS

L65-67: You could add a reference to support that first statement.

L68: Here you are referring to the IPCC report but I am not sure that it is the most relevant work to support the fact that eutrophication disrupts diversity, functionality, and NCP. Wouldn't it be more relevant to refer to the IPBES report here?

L82-85: Are Isbell et al. 2015 really showing that? As far as I understood, they are not looking at the relationships between resistance or recovery, and temporal invariability.

L85: Here you are speaking about temporal invariability of what?

L91-92: I think that you can remove '(resp. negatively)' and '(resp. decrease)' here. This idea is then developed in the sentence L92-93 so it is a bit redundant.

L98: I would remove 'Similarly'. It is another idea, not related to the paragraph before.

L101-103: Again, I am not sure that Scherber et al. 2010 are showing that? Could you please provide a relevant reference and explain the mechanisms involved in the relationship between the stability of species richness, composition and the stability of ecosystem functioning? There is also ample evidence that species asynchrony stabilizes biomass production (see e.g. Yachi & Loreau 1999, de Mazancourt et al. 2013, Craven et al. 2018). Indeed, a decrease in the productivity of some species can be compensated by an increase in the productivity of other species that are less affected by a disturbance or by environmental changes (Loreau & de Mazancourt 2013).

L113: You could explain a bit more what are these 'shifts in community composition', i.e. which plant functional strategies are favoured.

L119: Which facets of stability exactly increased with nutrient addition? Please expand a bit.

L121-125: This section is very superficial and would deserve a bit more explanation on the underlying mechanisms involved.

L144-145: These numbers are impressive but not very informative. It would be interesting to know how many dry, normal and wet seasons were recorded per site. Is it presented somewhere?

L153: In the Supplementary, I think you could also show the mean biomass and richness during the extreme event, during last extreme and one year after for the control and nutrient addition treatment.

L162-167: I think that the sensitivity analysis considering abundance-weighted diversity indices is interesting and could be more detailed. The effect sizes shown on Fig. S9 are actually changing whether you considered Q0, Q1 or Q2.

L181-183: The lower correlation between resistance and recovery under nutrient addition could be a spurious relationship. Could you justify how nutrient addition can affect the relationship between these two stability facets?

L195: Is the correlation between temporal invariability and resistance negative or positive? Please specify.

L197: When considering long-term data, the results can indeed change considerably. I think that this is a very important point that should be emphasized. In this study, you are using data on plant

community biomass and richness collected for 4 years but considering longer-term experiment might strongly affect your conclusion due to delayed responses of some plant species (see Lepš 2014).
L206-207: Very unclear sentence to me. Could you please provide more details?
L339: I would replace 'As the value of Q increases...' by 'The value of Q increases...', or rephrase that sentence because as such it is not very clear.

REFERENCES

- Craven, D., Eisenhauer, N., Pearse, W.D. et al. Multiple facets of biodiversity drive the diversity–stability relationship. *Nat. Ecol. Evol.* 2, 1579–1587 (2018). <https://doi.org/10.1038/s41559-018-0647-7>
- de Mazancourt, C. et al. Predicting ecosystem stability from community composition and biodiversity. *Ecol. Lett.* 16, 617–625 (2013).
- García-Palacios, P. et al. Climate mediates the biodiversity–ecosystem stability relationship globally. *Proc. Natl Acad. Sci. USA* 115, 8400–8405 (2018).
- Lepš, J. Scale-and time-dependent effects of fertilization, mowing and dominant removal on a grassland community during a 15-year experiment. *Journal of Applied Ecology*, 51, 978–987 (2014).
- Loreau, M., & De Mazancourt, C. Biodiversity and ecosystem stability: a synthesis of underlying mechanisms. *Ecol. Lett.* 16, 106–115 (2013).
- Ma, Z, et al. Climate warming reduces the temporal stability of plant community biomass production. *Nat. Comm.* 8:15378 (2017).
- Shi, Z, et al. Dual mechanisms regulate ecosystem stability under decade-long warming and hay harvest. *Nat. Comm.* 7:11973 (2016).
- Yachi, S. & Loreau, M. Biodiversity and ecosystem productivity in a fluctuating environment: the insurance hypothesis. *Proc. Natl Acad. Sci. USA* 96, 1463–1468 (1999).

First of all, we thank the reviewers and editors for the constructive comments and suggestions that, we believe, have strengthened our manuscript. We considered all suggestions and comments very seriously and adjusted our manuscript accordingly. Below, we give a detailed point-by-point response to the reviews. To clarify how and where we dealt with the suggestions, we added our responses (with numbering) right after the comments and suggestions (text in blue), and the actual changed text in the manuscript is indicated in red text, with their corresponding page and line numbers in the revised version (ms_ indicating changes).

Reviewer #1 (Remarks to the Author):

In this manuscript the authors examine a very impressive dataset of 55 grassland sites that are part of the Nutrient Network to see how nutrient additions influence several facets of stability. They find that nutrient additions tend to decrease temporal invariability and resistance to wet and dry extremes for species composition and richness, but not biomass. They also find that invariability is positively related to resistance across nutrient treatments and community aspects. These results are interesting and they provide useful information about how global changes interact to affect grassland plant community stability from several perspectives. The manuscript is well written, but I have several overarching concerns and comments that I believe would improve the manuscript.

1. We appreciate that the reviewer found our manuscript interesting and well written. We also appreciate the reviewer's comments and suggestions, which have been very helpful for us to improve the manuscript. Please see our responses below.

First, I think the manuscript would benefit greatly from showing some of the data directly, rather than simply summarizing correlations (Figs 2-4). I don't even think I saw any of the invariability, resistance, or recovery values reported in the supplementary materials, so it is very hard to get an idea of what the data look like. Specifically, it's not clear how invariable places are, whether they recover, etc. I understand that this could be a bit messy, and might need to be put in the supplement, but finding a way to show the values used in the correlations would improve the ability to interpret your results.

2. We appreciate these suggestions. We now present raw data for the five stability facets in three community aspects at each site in supplementary Fig. S6. We present the pairwise correlation among stability facets for three community aspects in Fig. S11 - Fig. S13. We present the pairwise correlation among stability of community aspects for five stability facets in Fig. S14.

“

Fig. S6. Five stability facets in three community aspects under ambient and nutrient addition treatment at each site. Panel title shows stability in one community aspect for a given stability facet. Temporal invariability in biomass and species richness were calculated after detrending. Stability facets in biomass and species richness were log-transformed. Resistance and recovery for all three community aspects were averaged over years to match the data structure of temporal invariability. A dot represents one block within a site, see Table S3 for site number for different stability facets. Dotted lines represent 1:1 line, dots fall above this line indicate nutrient addition increases stability, whereas dots fall below this line indicate nutrient addition decreases stability.”

“

Fig. S11. Pairwise correlation among stability facets in aboveground biomass at individual sites under control (red) and nutrient addition (blue). Resistance and recovery for biomass were averaged over years to match the data structure of temporal invariability. A dot represents one block within a site. Regression lines were fitted using `geom_smooth` with method of “lm” from the R package `ggplot` to illustrate the correlation within each treatment within each site. See Table S5 for number of sites available and overall estimate of the correlation.

Fig. S12. Pairwise correlation among stability facets in community composition at individual sites under control (red) and nutrient addition (blue). Resistance and recovery for community composition were averaged over years to match the data structure of temporal invariability. A dot represents one block within a site. Regression lines were fitted using `geom_smooth` with method of “lm” from the R package `ggplot` to illustrate the correlation within each treatment within each site. See Table S5 for number of sites available and overall estimate of the correlation.

Fig. S13. Pairwise correlation among stability facets in species richness at individual sites under control (red) and nutrient addition (blue). Resistance and recovery for species richness were averaged over years to match the data structure of temporal invariability. A dot represents one block within a site. Regression lines were fitted using `geom_smooth` with method of “lm” from the R package `ggplot` to illustrate the correlation within each treatment within each site. See Table S5 for number of sites available and overall estimate of the correlation.

Fig. S14. Pairwise correlation among stability of three community aspects for a given stability facet at individual sites under control (red) and nutrient addition (blue). A dot represents one block within a site. Regression lines were fitted using `geom_smooth` with method of “lm” from the R package `ggplot` to illustrate the pairwise correlation within each treatment within each site. See Table S6 for number of sites available and overall estimate of the correlation.

”

Second, the results and discussion section feels mostly like a results section, with only very shallow discussion included for your results. I think your manuscript could benefit greatly from more detailed interpretation of your results by putting the results into context and

elaborating on some of the statements that you provide. For example, you mention a few times that we should focus on increasing resistance rather than recovery in ecosystems. Can you provide any information about how you might do that?

3. We thank the reviewer for suggesting to give more detailed interpretation of our results. As suggested, we have substantially expanded the discussion and interpreted our results in greater depth. First, we interpreted the effects of nutrient addition on each stability facet in more detail (page 4, lines 173-219). Second, we discussed expected correlations between temporal invariability and resistance and recovery, what we observed, and implications (page 6, lines 243-249). In particular, we explained that we should focus on increasing resistance rather than recovery in ecosystems by using drought and flood resistance genotypes or species (page 6, lines 246-248). Third, we interpreted the discrepancy between our results of correlations among stability in community aspects for a given stability facet with the results from a meta-analysis and implications (page 6, lines 271-279). Fourth, we added a paragraph to address the robustness and limitations of our results (page 6, lines 281-299).

Third, I think you need stronger justification for focusing on wet and dry years. Could you also focus on hot years? Including more detail up front about why wet and dry years are especially interesting/important would be good.

4. We thank the reviewer for bringing this important point to our attention. We agree that hot years or temperature could also affect grassland plant species. Indeed, temperature and ground heat were included in the calculation of SPEI (standardized precipitation–evapotranspiration index), which was used to categorize dry and wet years. We explained this clearer in the methods (page 8, lines 364-367). Also, we justified the reason to focus on dry and wet growing seasons in the introduction. That is, drought and floods are increasing in intensity and frequency with generally negative effects on grassland ecosystems (page 2, lines 77-79; page 3, line 127-129).

“Potential Evapotranspiration is calculated using the Penman-Monteith formula taking into account the rate of change of saturation specific humidity with air temperature, net irradiance, ground heat flux, air temperature at 2 meters, wind speed at 2 meters, and vapor pressure deficit⁴⁹.”

“Meanwhile, climate extremes (e.g., droughts and floods) are increasing in both intensity and frequency, which can also have severe negative impacts on our society and ecosystems^{2,3}.”

“Nutrients and water availability are essential for the survival and growth of plants²⁵, an imbalance in their availability can have negative impacts on various community aspects²⁶.”

Fourth, some of your sites only have 4 years of data and it made me wonder whether that is enough to calculate stability? Many of the calculations rely on calculating a “baseline” or normal level, which seems like it would become increasingly sensitive to random variation

when you have few samples. It doesn't look like many sites have only four years, but it might be worth checking how sensitive your results are to time series length.

5. Thank you for this insightful suggestion. To test the robustness of our results to the experimental period, we re-performed our analyses using 22 sites that have run for at least 10 years (supplementary figures S21-S23). The main conclusions were generally consistent: (i) nutrient addition decreased the invariability and resistance to dry and wet growing seasons for both composition and richness (comparing figure 2 vs. figure S21); (ii) the majority of stability facet are uncorrelated, while nutrient addition changed the correlation for a few pairs of stability facets (comparing figure 3 vs figure S22, figure 4 vs figure S23). That said, there are some differences between results using 55 vs. 22 sites, e.g., the specific pairs of stability facets exhibiting significant correlations, due to differences in the number and identity of sites. In the revised manuscript, we have expanded the discussion by adding a paragraph to address the limitations of our analyses, including the test of robustness using 22 sites with ≥ 10 years data. Page 6, line 281-299.

“

Fig. S21. Effects of nutrient addition on each of the five stability facets in each of the three community three aspects. Compare with Figure 2 which used 55 sites with experimental years ranging from 4 to 15, here results were based on data from 22 sites with experimental years ranging from 10 to 15. Saturated line colors represent significant effects at $p \leq 0.05$, faded line colors represent non-significant effects.

Fig. S22. Pairwise correlations among five stability facets in three community aspects under ambient and nutrient addition conditions. Compare with Figure 3 which used 55 sites with experimental years ranging from 4 to 15, here results were based on data from 22 sites with experimental years ranging from 10 to 15. The significant effects (saturated colors) correspond to 95 % confidence intervals of a correlation coefficient does not overlap with 0.

Fig. S23. Pairwise correlations of stability among three community aspects under ambient and nutrient addition conditions. Compare with Figure 4 which used 55 sites with experimental years ranging from 4 to 15, here results were based on data from 22 sites with experimental years ranging from 10 to 15. The significant effects (saturated colors) correspond to 95 % confidence intervals of a correlation coefficient does not overlap with 0.”

“Robustness and limitations

To address the robustness of our results, we re-performed the analyses (i) using more extreme thresholds of SPEI to define dry and wet growing seasons (dry: ≤ 10 th

percentile; wet: ≥ 90 th percentile; Fig. S15-Fig. S17); (ii) after removing the long-term linear trends of SPEI (Fig. S18-Fig. S20); and (iii) based on 22 sites with at least 10-year observations (Fig. S21-Fig. S23). These analyses led to similar patterns as those presented above: (i) nutrient addition decreased the temporal invariability and resistance during dry and wet growing seasons for both composition and richness, but not for biomass; (ii) the majority of pairs of stability measures were uncorrelated, while nutrient addition changed the correlations for a few pairs of stability measures. That said, the specific pairs of stability measures that exhibit significant correlations do change due to differences in the number and identity of sites included in these additional analyses. One limitation of our analyses is to appropriately quantify resistance and recover in the presence of consecutive climate extremes (e.g. dry and dry, dry and wet, or wet and wet growing seasons). Existing theories mostly consider a single pulse perturbation to quantify resistance and recovery (e.g., Fig. 1), making it challenging to disentangle ecosystem responses to consecutive extreme growing seasons. In such cases, a previous dry (wet) growing season may affect the community's response to the subsequent wet (dry) growing seasons, i.e., legacy effects^{42,43}. New theory and methodology are needed to address ecosystem responses to consecutive and repeated perturbations.”

Finally, the main message of your manuscript seems to be that invariability and resistance are related, and not related to recovery, which seems like it should be that way based on how the metrics are calculated (higher resistance is less deviation from the mean, which is also essentially invariability, and where there is recovery, there cannot be invariability and resistance must be relatively low). So, I think reporting those results are not especially profound. I think it might be more interesting to focus on where these expected relationships are not so obvious and why that might be.

6. Thank you for these insightful comments. We agree that compared to the positive invariability-resistance relationships, the more important finding of the paper is the weak relationships among different stability measures (i.e., high dimensionality of stability), e.g., those between invariability and recovery for each community aspect or between stability of different community aspects. We realized that the confusion could be induced by emphasizing the invariability-resistance relationships in the abstract of the original manuscript. In the revised manuscript, we have weakened the emphasis on the invariability-resistance relationship (e.g., removing it from the abstract and reducing related discussion) and expanded the discussion on the weak relationships between stability measures: both the reasons and implications (page 5, line 238-249; page 6, line 270-279).

“Notably, temporal invariability was **positively** correlated with resistance, but not recovery, under both treatments **for** all three community aspects **investigated** (Fig. 3; see **Fig. S11- Fig. S13 for correlations at individual sites**). Our result is in line with findings based on manipulated biodiversity experiments³⁸, which shows that grassland species richness increases temporal invariability by enhancing resistance rather than recovery. **Combined, these results indicate that in spite of the expected positive associations between temporal invariability with both resistance and recovery by definition and quantification**, long-term temporal invariability of different community aspects relies on

their short-term responses during climate extremes rather than their **subsequent** recovery. Thus, strategies **aimed at enhancing** community resistance, such as **using drought and flood resistant genotypes or species**, are also **crucial** for **promoting** the temporal invariability of ecosystem properties in eutrophic conditions.”

“Our findings differ from results from a recent meta-analysis showing that compositional stability and biomass stability are positively correlated¹⁰. **This discrepancy may be understood from the different definitions of stability facets and correlations. For instance, while our study calculated correlations among stability measures within experimental sites, the meta-analysis derived correlations across sites along environmental gradients which might mediate positive correlations between stability measures.** Our results suggest that **stability of species diversity, community composition, and biomass** represent separate dimensions. **Thus, different and** context-dependent strategies may be required for effective management **for different community aspects.**”

In addition, here are a few line by line comments:

Abstract:

Line 50-51: The phrasing here makes it hard to pick out the 5 dimensions. Maybe make it a numbered list in the parentheses?

7. Thanks for the suggestion. We have revised the abstract to clarify the 5 stability facets in 3 community aspects that we investigated. Page 2, line 59-62.

“five facets of stability (temporal invariability, **resistance during dry and wet growing seasons, recovery after dry and wet growing seasons**) measured on three community aspects (aboveground biomass, community composition, and species richness).”

Line 55-56: Is this not somewhat mathematically expected, as resistance should be related to invariability because both are a lack of change through time?

8. As explained above (response #7), we have reduced emphasis on this result and have deleted this sentence from the abstract.

Line 57-58: I’m not sure why this follows from the previous sentence. Are you trying to say that the grasslands that you studied did not show recovery after dry and wet years? It still seems like recovery should be an important element of stability.

9. Sorry for the confusion. Our results showed that adding nutrients generally did not change recovery (Fig. 3; newly added figure with effects at individual sites in Fig. S6), rather than that our study systems did not show recovery. We fully agree that recovery is an important element of stability. In the revised manuscript, we have removed this sentence to highlight the overall high dimensionality of stability in the abstract. Page 2, line 64-66.

“Different stability measures were largely uncorrelated under both ambient and eutrophic conditions, indicating consistently high dimensionality.”

Introduction:

Lines 65-67: Citation for this statement?

10. Added. Page 2, Line 73-75.

“In 2020, the Convention on Biological Diversity reported that only 8% of the world's nations met the target of limiting excess nutrients to a level that is not detrimental to ecosystem functioning¹.”

Line 67: Diversity and functionality of what?

11. We now clarify that we are referring to natural ecosystems. Page 2, Line 75-77.

“This failure means that eutrophication, which disrupts diversity, functionality of many ecosystems, including grasslands, and nature's contributions to humanity^{2,3}, could threaten our long-term survival and prosperity.”

Line 91-93: Not sure what you mean by “resp. negatively”, as you later say that if facets are negatively correlated, then you can't optimize both.

12. Sorry for the confusion. In the revised manuscript, we deleted “resp. negatively”.

Lines 100-103: I think providing more justification for picking these three community aspects would be useful, especially to set up expectations about which aspects should be more or less stable and which should be correlated and how.

13. Thanks for this suggestion. We have added more justification in the Introduction to explain why we focus on these three community aspects. page 3, line 109-125.

“While most studies have focused on the stability of aboveground biomass, the stability of other community aspects such as community composition and species richness may also be essential for regulating ecosystem functions and maintaining ecosystem stability. For instance, higher invariability in community composition usually leads to higher invariability in aboveground biomass in grasslands¹⁷⁻¹⁹. But low stability in community composition (large compositional variation) can also be associated with high biomass stability if different species exhibit compensatory dynamics over time^{20,21}. Higher species richness (often the mean) has been shown to enhance temporal invariability of aboveground biomass because population decreases in some species due to climate extremes may be compensated by increases in other species that can endure harsh climate conditions²². This relationship can be modulated by global change factors such as eutrophication and aridity^{23,24}. Moreover, species richness itself is likely to vary over time

under climate change, and the correlations among stability in species richness and that in aboveground biomass or community composition is largely unclear. A simultaneous understanding of the various facets of stability in multiple community aspects is crucial for predicting and managing ecosystem stability in the face of global environmental change such as eutrophication.”

Results and discussion:

No specific comments, see overarching comments above.

Methods:

Lines 279-280: Is it potentially problematic if your sites have directional trends in SPEI from 1901-2021? It seems like it may be a problem because if you have a positive trend through time, you will be more likely to detect extreme wet years at the end of the time series. Did you check this by detrending the SPEI data?

14. Thank you for bringing this important point to our attention. You are right that a positive trend through time would lead to higher likelihood to detect extreme wet years at the end of the time series. Similarly, a negative trend through time would lead to higher likelihood to detect extreme dry years at the end of the time series. We incorporated this in our manuscript (methods; page 10, line 448-451). Following your suggestion, we tested the temporal trend of SPEI for each site and found that 21 out of 55 sites showed significant, though weak, linear trends in SPEI. We then reanalyzed the data after detrending SPEI. Our re-analyses showed overall similar results to those presented in the main text, except that the correlations between some pairs of stability facets within each community aspect changed from significant to non-significant. See Fig. S18-Fig. S20 in supplementary file. In the revised manuscript, we have included these results in the subsection “Robustness and limitations” in the discussion (page 6 lines 281-291).

“Second, we tested the temporally linear trend of SPEI for each site and re-analyzed the data after removing the trends of SPEI, because a positive (negative) trend of SPEI would lead to higher likelihood to detect extreme wet (dry) years at the end of the time series.”

“

Fig. S18. Effects of nutrient addition on each of the five stability facets in each of the

three community three aspects. Compare with Figure 2, here dry and wet growing seasons were identified after detrending SPEI. That is, we used linear regression (function “lm”) to fit SPEI over time (from 1902 to 2021), we then used the residuals from this model as the detrended SPEI. Saturated line colors represent significant effects at $p \leq 0.05$, faded line colors represent non-significant effects.

Fig. S19. Pairwise correlations among five stability facets in three community aspects under control and nutrient addition conditions. Compare with Figure 2, here Dry and wet growing seasons were identified after detrending SPEI. That is, we used linear regression (function “lm”) to fit SPEI over time (from 1902 to 2021), we then used the residuals from this model as the detrended SPEI. The significant effects (saturated colors) correspond to 95 % confidence intervals of a correlation coefficient does not overlap with 0.

Fig. S20. Pairwise correlations of stability among three community aspects under control and nutrient addition conditions. Compare with Figure 2, here Dry and wet growing seasons were identified after detrending SPEI. That is, we used linear regression (function “lm”) to fit SPEI over time (from 1902 to 2021), we then used the residuals from

this model as the detrended SPEI. The significant effects (saturated colors) correspond to 95 % confidence intervals of a correlation coefficient does not overlap with 0. ”

“*Robustness and limitations*”

To address the robustness of our results, we re-performed the analyses (i) using more extreme thresholds of SPEI to define dry and wet growing seasons (dry: \leq 10th percentile; wet: \geq 90th percentile; Fig. S15-Fig. S17); (ii) after removing the long-term linear trends of SPEI (Fig. S18-Fig. S20); and (iii) based on 22 sites with at least 10-year observations (Fig. S21-Fig. S23). These analyses led to similar patterns as those presented above: (i) nutrient addition decreased the temporal invariability and resistance during dry and wet growing seasons for both composition and richness, but not for biomass; (ii) the majority of pairs of stability measures were uncorrelated, while nutrient addition changed the correlations for a few pairs of stability measures. That said, the specific pairs of stability measures that exhibit significant correlations do change due to differences in the number and identity of sites included in these additional analyses.”

There is not enough information about how the resistance and recovery metrics were calculated. For example, I don't understand why invariability and resistance should range between -1 and 0 and not 0 to 1. It also seems theoretically possible that resistance could be greater than 1 (see aboveground biomass for a wet year in figure 1). I may be misunderstanding something, but I think this needs more clarity.

15. Thanks for raising this matter. We explained more clearly how stability facets were calculated. Temporal invariability and resistance for community composition should range from 0 to 1, because we calculated them using 1- Bray–Curtis dissimilarity (range from 0 to 1). “between -1 and 0” in the previous version was a mistake by writing. Page 4, line 158-162; page 9, line 399-426.

“To enable comparison among sites with varying conditions, we quantified resistance as the inverse of the proportional deviation of a community aspect from its normal levels during a dry or wet growing season. We quantified recovery as the ratio of deviation in a community aspect during to that after a dry or wet growing season following ref³⁸.”

“Methods used for quantifying stability facets in aboveground biomass and species richness are illustrated in Fig. 1. Specifically, temporal invariability was calculated as $\frac{\mu}{\sigma}$, where μ is the mean in aboveground biomass or species richness over the experimental years. σ is the standard deviation of aboveground biomass or species richness over time, which was calculated after detrending to remove variation due to directional change over time. That is, we first used linear regression (function “lm”) to fit species richness or biomass against experimental years for each subplot, we then used the residuals from this model to calculate the standard deviation. Following ref³⁸, resistance and recovery were calculated as $\frac{\bar{Y}_n}{|Y_e - \bar{Y}_n|}$ and $\frac{|Y_e - \bar{Y}_n|}{|Y_{e+1} - \bar{Y}_n|}$, where \bar{Y}_n , Y_e , and Y_{e+1} are aboveground biomass or species richness during normal growing seasons (average over all normal growing seasons), during an extreme, and one year after an extreme growing season. Stability facets in biomass and species richness were log-transformed to improve homogeneity of variance. The composition-related facets of stability were calculated using Bray–Curtis

dissimilarity metric based on cover data^{50,51}. Values of Bray-Curtis dissimilarity ranges from 0 to 1, with higher dissimilarity being closer to 1. We used similarity (i.e., 1 - dissimilarity) to measure stability in community composition, with higher values indicating higher stability. Temporal invariability was calculated as the overall community similarity over all experimental years using the function “beta.multi.abund” from the R package betapart⁵⁰. The average cover for all species during normal growing seasons under each treatment within a block was constructed as a reference community for calculating resistance and recovery. Resistance was calculated as the similarity of the plant community under an extreme growing season compared with the reference (values ranged from 0 to 1) using the R function “beta.pair.abund”. Similarly, we calculated similarity of the plant community one year after an extreme growing season compared with the reference. Recovery was then calculated as the ratio of similarity of the community one year after to that during an extreme growing season (ranged from 0 to inf). Resistance and recovery for all three community aspects were averaged over years to match the data structure of temporal invariability. For all stability facets in the three community aspects, higher values represent higher stability.”

Figures:

Figure 1: This is very helpful for visualizing the stability facets. Nice job.

16. Thank you.

Reviewer #2 (Remarks to the Author):

SUMMARY

The authors have conducted an interesting study in which they investigate how different facets of ecological stability, the resistance during and the recovery after extreme climate events of plant communities are affected by nutrient addition in grasslands. Using data collected in 55 grassland sites during at least 4 years of nutrient addition experiments (NutNet experiments), they test how nutrient addition affects the correlation between the stability, the resistance during and the recovery after dry and wet growing seasons of three plant community aspects: aboveground biomass, the composition of the plant community and its species richness. They show that nutrient addition reduces the stability and resistance of species richness and community composition during dry and wet climate events. However, the stability and resistance of aboveground biomass was not affected by nutrient addition. The stability and the resistance of the different community aspects were positively correlated - independently of nutrient addition - but it was not correlated with recovery.

Overall, I think that these results provide useful information about the consequences of two important global change drivers for ecosystem functioning: nutrient addition and climate changes. I have only few minor concerns with that manuscript:

17. We are pleased that you found our manuscript and analyses useful. We appreciate your helpful comments and please find our responses below.

1. I was wondering why the authors used the term ‘temporal invariability’ and not the term ‘stability’? I think that the term ‘stability’ is more often used and understandable by a wide readership. I would therefore consider to homogenizing the wording and using this term instead of ‘temporal invariability’.

18. We agree that “stability” has been mostly used in the literature to represent the temporal invariability as defined in our study. However, we would like to emphasize that there has been a shift in recent years toward the use of “invariability”. That shift has been motivated by the fact that “stability” is a multi-faceted concept, which includes resistance, recovery, temporal invariability, etc. For instance, resistance and recovery represent ecosystem responses to a pulse perturbation in relative short terms, whereas invariability represents the response in relatively long terms in the face of multiple perturbations. In other words, invariability represents one facet, the most commonly used, to quantify stability. Therefore, and in line with this shift in the literature and the clear separation of stability as a concept and invariability as a metric, we would like to retain our initial choice. In the revised manuscript, we have added sentences to justify this term. Page 2, line 85-90.

“Traditionally, stability has been assessed through temporal invariability, which involves calculating the mean of an ecosystem property divided by its standard deviation⁴⁻⁶. As a result, temporal invariability has been commonly referred to as temporal stability. In this paper, as we consider multiple facets of stability (including invariability) to quantify ecosystem responses through time, we use invariability to avoid confusion.”

2. The references cited are not always adapted to support the ideas developed, especially in the ‘Introduction’ section (see specific suggestions below). I think that the authors should check carefully that they are using appropriate references.

19. Thanks for raising this important point. We have now checked the references throughout the text to make sure that they fit to the context. In particular, following your suggestions, we cited the IPBES report (page 2, line 77), deleted the reference Scherber et al. 2010.

3. I was left a bit disappointed with the ‘Results and discussion’ section. I am aware that the authors are limited by the word count but I think that the discussion points could be expanded. For now, the discussion is quite shallow (see also specific suggestions).

20. We thank the reviewer for suggesting to give more detailed interpretation of our results. As suggested, we have substantially expanded the discussion and interpreted our results in greater depth. First, we interpreted the effects of nutrient addition on each stability facet in more detail (page 4, lines 173-219). Second, we discussed expected correlations between temporal invariability and resistance and recovery, what we observed, and implications (page 6, lines 243-249). In particular, we explained that we should focus on

increasing resistance rather than recovery in ecosystems by using drought and flood resistance genotypes or species (page 6, lines 246-248). Third, we interpreted the discrepancy between our results of correlations among stability in community aspects for a given stability facet with the results from a meta-analysis and implications (page 6, lines 271-279). Fourth, we added a paragraph to address the robustness and limitations of our results (page 6, lines 281-299).

4. Finally, in this study, the authors look at how nutrient addition affects the relationship between different stability facets, including the resistance during and recovery after a dry or wet event. I think it is also important to keep in mind and acknowledge that climatic conditions can modulate the relationships between biodiversity and ecosystem functioning (see e.g. Shi et al. 2016, Ma et al. 2017, Garcia-Palacios et al. 2018).

21. Thanks for the suggestion. We read the three manuscripts suggested by the review carefully, these three manuscripts investigated temporal invariability measured as mean/sd. Garcia-Palacios et al. (2018) found that the relationship between species richness and temporal invariability of NDVI was modulated by aridity gradient in global drylands. Ma et al. (2017) found that warming decreased temporal invariability, but temporal invariability was not influenced by species richness in an experiment manipulating warming and precipitation in an alpine grassland in Tibetan Plateau. In Fig. 3b of their paper, the authors show that species richness and temporal invariability in biomass is positively correlated across treatments. Shi et al. (2016) found that warming increased temporal invariability, and species richness also explained part of the variation in temporal invariability of biomass production in an experiment manipulating warming and cutting in the Kessler Atmospheric and Ecological Field Station (KAEFS). In Fig. 2b of their paper, the authors show temporal invariability and biodiversity (the first component of a principal component analysis with richness, evenness and diversity) is positively correlated across treatments. In the revised manuscript, we added a sentence to acknowledge the point that global change factors including climate conditions may modulate the relationship between species richness and temporal invariability. Page 3, line 116-120. Additionally, we added a sentence to acknowledge that climate conditions may modulate the relationship between stability facets. Page 3, line 93-97.

“Higher species richness (often the mean) has been shown to enhance temporal invariability of aboveground biomass because population decreases in some species due to climate extremes may be compensated by increases in other species that can endure harsh climate conditions²². This relationship can be modulated by global change factors such as eutrophication and aridity^{23,24}.”

“A few studies (mostly single-site experiments) exploring multiple facets of stability have showed that different stability facets are generally uncorrelated, and these correlations or lack thereof may differ under global change factors including climate extremes, eutrophication, consumer removal, light addition, and heatwaves^{8,10-16}.”

Below, I present more specific suggestions that may help, I hope, to improve the manuscript.

SPECIFIC SUGGESTIONS

L65-67: You could add a reference to support that first statement.

22. Added. Page 2, Line 73-75.

“In 2020, the Convention on Biological Diversity reported that only 8% of the world's nations met the target of limiting excess nutrients to a level that is not detrimental to ecosystem functioning¹.”

L68: Here you are referring to the IPCC report but I am not sure that it is the most relevant work to support the fact that eutrophication disrupts diversity, functionality, and NCP. Wouldn't it be more relevant to refer to the IPBES report here?

23. Thank you for the suggestion. We agree that the IPBES report is more relevant, and we now cite this reference. Page 2, Line 75-77.

“This failure means that eutrophication, which disrupts diversity, functionality of many ecosystems, including grasslands, and nature's contributions to humanity^{2,3}, could threaten our long-term survival and prosperity.”

L82-85: Are Isbell et al. 2015 really showing that? As far as I understood, they are not looking at the relationships between resistance or recovery, and temporal invariability.

24. Indeed, Isbell et al. (2015) did not directly look at the relationships between temporal invariability and resistance, temporal invariability and recovery. Isbell et al. (2015) looked at the relationships between species richness and each of the three stability facets mentioned above. He found that species richness increases temporal invariability and resistance, rather than recovery. We agree that this is indirect evidence for this argument. In the revised manuscript, we removed this sentence from the Introduction, but adjusted and incorporated it in the discussion. Page 5, line 240-242.

“Our result is in line with findings based on manipulated biodiversity experiments³⁸, which infers that grassland species richness increases temporal invariability by enhancing resistance rather than recovery.”

L85: Here you are speaking about temporal invariability of what?

25. We are referring to the temporal invariability of aboveground biomass. We removed this sentence from the text to focus on the general correlations among stability facets.

L91-92: I think that you can remove ‘(resp. negatively)’ and ‘(resp. decrease)’ here. This idea is then developed in the sentence L92-93 so it is a bit redundant.

26. Thanks. We deleted these words.

L98: I would remove ‘Similarly’. It is another idea, not related to the paragraph before.

27. Thanks. We removed it.

L101-103: Again, I am not sure that Scherber et al. 2010 are showing that? Could you please provide a relevant reference and explain the mechanisms involved in the relationship between the stability of species richness, composition and the stability of ecosystem functioning?

There is also ample evidence that species asynchrony stabilizes biomass production (see e.g. Yachi & Loreau 1999, de Mazancourt et al. 2013, Craven et al. 2018). Indeed, a decrease in the productivity of some species can be compensated by an increase in the productivity of other species that are less affected by a disturbance or by environmental changes (Loreau & de Mazancourt 2013).

28. Thank you for this important note. We agree that Scherber et al. (2010) is not the most relevant reference here and have deleted it. Instead, we have added several sentences to justify the importance of understanding stability in community composition and species richness and their relationships with stability of community biomass. page 3, line 109-123.

“While most studies have focused on the stability of aboveground biomass, the stability of **other** community aspects such as community composition and species richness **may also** be essential for regulating ecosystem functions and maintaining ecosystem stability. For instance, **higher invariability in community composition usually leads to higher invariability in aboveground biomass in grasslands¹⁷⁻¹⁹. But low stability in community composition (large compositional variation) can also be associated with high biomass stability if different species exhibit compensatory dynamics over time^{20,21}. Higher species richness (often the mean) has been shown to enhance temporal invariability of aboveground biomass because population decreases in some species due to climate extremes may be compensated by increases in other species that can endure harsh climate conditions²². This relationship can be modulated by global change factors such as eutrophication and aridity^{23,24}. Moreover, species richness itself is likely to vary over time under climate change, and the correlations among stability in species richness and that in aboveground biomass or community composition is largely unclear. A simultaneous understanding of the various facets of stability in multiple community aspects is crucial for predicting and managing ecosystem stability in the face of global environmental change **such as eutrophication.**”**

L113: You could explain a bit more what are these ‘shifts in community composition’, i.e. which plant functional strategies are favoured.

29. We now added some words to explain what we meant. Page 3, line 134-136.

“Furthermore, eutrophication has been well documented to decrease species richness and cause vegetation shifts toward **domination by fast-growing and invasive species^{35,36}.**”

L119: Which facets of stability exactly increased with nutrient addition? Please expand a bit.

30. Polazzo and Rico (2021) found that nutrient addition increased functional recovery and resilience, leading to lower dimensionality of functional stability (measured by abundance). In the revised manuscript, we deleted this sentence, we revised this paragraph to focus on how adding nutrients may alter correlations among stability facets. Page 3, line 140-151.

“On the one hand, eutrophication can enhance interspecific competition and deterministic community assembly processes, thus strengthening the correlation among different stability facets^{11,12}. But on the other hand, eutrophication may promote stochastic community assembly via increasing soil fertility and productivity, potentially weakening the correlations among different stability facets³⁷. Indeed, different facets of stability measured in different community aspects may respond differentially (in both direction and magnitude) to eutrophication^{18,31,35}, leading to either weakened or strengthened correlations among stability measures. However, a recent study in a grassland finds that eutrophication does not alter relationships among stability in community biomass and composition¹⁸. Overall, a systematic investigation of the correlations among stability facets in various community aspects and their responses to eutrophication is still lacking.”

L121-125: This section is very superficial and would deserve a bit more explanation on the underlying mechanisms involved.

31. We have expanded the text to explain the possible mechanisms involved. Please see our response 30.

L144-145: These numbers are impressive but not very informative. It would be interesting to know how many dry, normal and wet seasons were recorded per site. Is it presented somewhere?

32. This information can be found in Figs. S2 and S5. We have specified this in the revised main text. Page 4, line 166-168.

“In total, we recorded 150 dry, 247 normal, and 131 wet growing seasons across all sites during the study period (see Fig. S2 for the number of dry and wet growing seasons at individual sites).”

L153: In the Supplementary, I think you could also show the mean biomass and richness during the extreme event, during last extreme and one year after for the control and nutrient addition treatment.

33. Thanks for the suggestion. We show these results in Fig. S7 to Fig. S9.

Fig. S7. Change in aboveground biomass (g m^{-2} ; upper panel) and the magnitude of biomass deviation from normal levels (lower panel) under control and nutrient addition treatments. The normal levels are the means of aboveground biomass over normal growing seasons within treatments in each block at each site. During normal growing seasons, on average, nutrient addition increased aboveground biomass by 48% (control: 318.54; nutrient addition: 471.44; g.m^{-2}). Change in aboveground biomass refers to biomass differences from normal levels, thus values can be positive or negative. Deviation in aboveground biomass refers to absolute change in biomass from normal levels, thus values are positive only. “During the last extreme” refers to the last dry or wet growing season when more than one dry or wet growing seasons occur consecutively. “One year after” refers to a normal growing season after a dry or wet growing season. “During the last” and “one year after” were used to calculate recovery. Dots indicate average values over dry or wet growing seasons across sites. Error bars are 95% bootstrapped confidence intervals.

Fig. S8. Average community similarity during normal growing seasons, during and one year after dry and wet growing seasons under control and nutrient addition treatments. “During the last extreme” refers to the last dry or wet growing season when more than one dry or wet growing seasons occur consecutively. “One year after” refers to a normal growing season after a dry or wet growing season. “During the last” and “one year after” were used to calculate recovery. Dots are means, error bars are 95% bootstrapped confidence intervals.

Fig. S9. Change in species richness (spp m⁻²; upper panel) and the magnitude of richness deviation from normal levels (lower panel) under control and nutrient addition treatments. The normal levels are the means of species richness over normal growing seasons within treatments in each block at each site. During normal growing seasons, on average, nutrient addition decreased species richness by 19% (control: 12.18; nutrient addition: 9.82; spp.m⁻²). Change in species richness refers to richness difference from normal levels, thus values can be positive or negative. Deviation in species richness refers to absolute

change in richness from normal levels, thus values are positive only. “During the last extreme” refers to the last dry or wet growing season when more than one dry or wet growing seasons occur consecutively. “One year after” refers to a normal growing season after a dry or wet growing season. “During the last” and “one year after” were used to calculate recovery. Dots indicate average values during dry or wet growing seasons and across sites. Error bars and thin lines are 95% bootstrapped confidence intervals.”

L162-167: I think that the sensitivity analysis considering abundance-weighted diversity indices is interesting and could be more detailed. The effect sizes shown on Fig. S9 are actually changing whether you considered Q0, Q1 or Q2.

34. This figure is now Fig. S10 in the revised version. Indeed, effect size of nutrient addition on resistance during wet growing seasons decreased as Q increased from 0 to 2. We clarified this point in the discussion. Page 5, line 205-211.

“We found that nutrient addition similarly reduced invariability and resistance of cover-weighted species diversity (e.g., Hill number equals 0, 1, and 2) during dry growing seasons. But the effects of nutrient addition on resistance during wet growing seasons decreased with increasingly high weights for abundant species (non-significant for Hill number equal to 2; Fig. S10). This suggests that dominant plant species may be more resistant than rarer species during wet growing seasons under eutrophication (Fig. S10).”

L181-183: The lower correlation between resistance and recovery under nutrient addition could be a spurious relationship. Could you justify how nutrient addition can affect the relationship between these two stability facets?

35. Thank you for this comment. We checked the results and found that the correlation between resistance and recovery (for biomass during dry growing seasons) shifted from -0.29 (-0.58~0.00; 95% confidence interval) under ambient conditions to -0.25 (-0.54~0.04; 95% confidence interval) under nutrient addition. So, we agree with your concern that this result may reflect the artificial choice of significance level (95% confidence CIs do not overlap with 0). In the revised manuscript, we weakened the statement on this result. Page 5, line 232-233.

“For instance, nutrient addition **weakened** the negative correlation between resistance during dry and wet growing seasons in biomass.”

L195: Is the correlation between temporal invariability and resistance negative or positive? Please specify.

36. The correlation is positive. We now specify this in the main text. Page 5, line 238-240.

“Notably, temporal invariability was **positively** correlated with resistance, but not recovery, under both treatments **for** all three community aspects **investigated** (Fig. 3; see Fig. S11- Fig. S13 for correlations at individual sites).”

L197: When considering long-term data, the results can indeed change considerably. I think that this is a very important point that should be emphasized. In this study, you are using data on plant community biomass and richness collected for 4 years but considering longer-term experiment might strongly affect your conclusion due to delayed responses of some plant species (see Lepš 2014).

37. Thank you for this insightful suggestion. To test the robustness of our results to the experimental period, we re-performed our analyses using 22 sites that have run for at least 10 years (supplementary figures S21-S23). The main conclusions were generally consistent: (i) nutrient addition decreased the invariability and resistance to dry and wet growing seasons for both composition and richness (comparing figure 2 vs. figure S21); (ii) the majority of stability facet are uncorrelated, while nutrient addition changed the correlation for a few pairs of stability facets (comparing figure 3 vs figure S22, figure 4 vs figure S23). That said, there are some differences between results using 55 vs. 22 sites, e.g., the specific pairs of stability facets exhibiting significant correlations, due to differences in the number and identity of sites. In the revised manuscript, we have expanded the discussion by adding a paragraph to address the limitations of our analyses, including the test of robustness using 22 sites with ≥ 10 years data. Page 6, line 281-292.

“

Fig. S21. Effects of nutrient addition on each of the five stability facets in each of the three community three aspects. Compare with Figure 2 which used 55 sites with experimental years ranging from 4 to 15, here results were based on data from 22 sites with experimental years ranging from 10 to 15. Saturated line colors represent significant effects at $p \leq 0.05$, faded line colors represent non-significant effects.

Fig. S22. Pairwise correlations among five stability facets in three community aspects under ambient and nutrient addition conditions. Compare with Figure 3 which used 55 sites with experimental years ranging from 4 to 15, here results were based on data from 22 sites with experimental years ranging from 10 to 15. The significant effects (saturated colors) correspond to 95 % confidence intervals of a correlation coefficient does not overlap with 0.

Fig. S23. Pairwise correlations of stability among three community aspects under ambient and nutrient addition conditions. Compare with Figure 4 which used 55 sites with experimental years ranging from 4 to 15, here results were based on data from 22 sites with experimental years ranging from 10 to 15. The significant effects (saturated colors) correspond to 95 % confidence intervals of a correlation coefficient does not overlap with 0.”

“Robustness and limitations

To address the robustness of our results, we re-performed the analyses (i) using more extreme thresholds of SPEI to define dry and wet growing seasons (dry: ≤ 10 th

percentile; wet: ≥ 90 th percentile; Fig. S15-Fig. S17); (ii) after removing the long-term linear trends of SPEI (Fig. S18-Fig. S20); and (iii) based on 22 sites with at least 10-year observations (Fig. S21-Fig. S23). These analyses led to similar patterns as those presented above: (i) nutrient addition decreased the temporal invariability and resistance during dry and wet growing seasons for both composition and richness, but not for biomass; (ii) the majority of pairs of stability measures were uncorrelated, while nutrient addition changed the correlations for a few pairs of stability measures. That said, the specific pairs of stability measures that exhibit significant correlations do change due to differences in the number and identity of sites included in these additional analyses.”

L206-207: Very unclear sentence to me. Could you please provide more details?

38. Sorry for the confusion. We clarified this point in the revised text. Page 6, line 263-268.

“We found that for each stability facet, the correlations **among** the stability of the three community aspects were generally weak, under both ambient and nutrient addition (Fig. 4; see Fig. S14 for correlations at individual sites; Table S6). The consistently **weak correlations among stability of aboveground biomass, species richness, and community composition may indicate** differential responses of different community aspects to climatic fluctuations (Fig. 2).”

L339: I would replace ‘As the value of Q increases...’ by ‘The value of Q increases...’, or rephrase that sentence because as such it is not very clear.

39. Thanks. We rephrased it as suggested. Page 9, line 435-437.

“we calculated effective species diversity corresponding to Hill numbers Q ranging from 0 to 2 (**an increase in Q indicating greater weights of abundant species**)⁵⁴.”

REVIEWERS' COMMENTS

Reviewer #1 (Remarks to the Author):

Thank you for your very thorough work in revising your manuscript. I believe that it is very much improved and I appreciate your attention to the reviewer comments.

Reviewer #2 (Remarks to the Author):

The authors have very well addressed all the comments raised in the last review. Following recommendations, the authors have now (i) carefully revised the references and they are now citing more appropriate studies; (ii) expanded the 'Results and discussion' section; (iii) added very informative sensitivity analyses.

I have no further comment and I think that this manuscript is an important contribution.